

# Non-linear curve adjustments widen biological interpretation of relative growth analyses of the clam *Tivela mactroides* (Bivalvia, Veneridae)

Alexander Turra[1], Guilherme N. Corte[1,2], Antonia Cecília Z. Amaral[2], Leonardo Q. Yokoyama[1,3] and Márcia R. Denadai[2]

[1] Departamento de Oceanografia Biológica / Instituto Oceanográfico, Universidade de São Paulo, São Paulo, SP, Brasil
[2] Departamento de Biologia Animal / Instituto de Biologia, Universidade Estadual de Campinas, Campinas, SP, Brazil
[3] Departamento de Ciências do Mar / Instituto Saúde e Sociedade, Universidade Federal de São Paulo, Santos, SP, Brazil

Corresponding author
Alexander Turra, turra@usp.br

## ABSTRACT

Evaluation of relative (allometric) growth provides useful information to understand the development of organisms, as well as to aid in the management of fishery-exploited species. Usually, relative growth analyses use classical models such as the linear equation or the power function (allometric equation). However, these methods do not consider discontinuities in growth and may mask important biological information. As an alternative to overcome poor results and misleading interpretations, recent studies have suggested the use of more complex models, such as non-linear regressions, in conjunction with a model selection approach. Here, we tested differences in the performance of diverse models (simple linear regression, power function, and polynomial models) to assess the relative growth of the trigonal clam *Tivela mactroides*, an important fishing resource along the South American coast. Regressions were employed to relate parameters of the shell (length ($L$), width ($W$), height ($H$) and weight ($SW$)) among each other and with soft parts of the organism (dry weight (DW) and ash-free dry weight ($ASDW$)). Then, model selection was performed using the information theory and multi-model inference approach. The power function was more suitable to describe the relationships involving shell parameters and soft parts weight parameters (i.e., $L$ vs. $SW$, $DW$, and $AFDW$, and $SW$ vs. $DW$). However, it failed in unveiling changes in the morphometric relationships between shell parameters (i.e., $L$ vs. $W$ and $H$; $W$ vs. $H$) over time, which were better described by polynomial functions. Linear models, in turn, were not selected for any relationship. Overall, our results show that more complex models (in this study polynomial functions) can unveil changes in growth related to modifications in environmental features or physiology. Therefore, we suggest that classical and more complex models should be combined in future studies of allometric growth of molluscs.

## INTRODUCTION

Relative or allometric growth analysis allows a detailed evaluation of the proportionality among the body parameters of individuals, and is useful to estimate the production and biomass of a population from a single parameter such as length or width (*Rainer, 1985*; *Urban & Campos, 1994*; *Vasconcelos et al., 2018*). It also provides a better comprehension of modifications in life strategies of species unraveling important steps in their development (*Katsanevakis et al., 2006*; *Rabaoui et al., 2007*). Changes in growth rates are usually linked to changes in environmental features (e.g., food availability), physiology (e.g., gonad maturation and spawning periods), or biotic interactions (e.g., increase of competition or predation), and must be considered when evaluating morphological modifications during the ontogeny of the individuals (*Rabaoui et al., 2007*; *Caill-Milly et al., 2014*). The analysis of allometric growth is therefore an important tool to support exploitation and management of harvested species (e.g., improving the size-selectivity of fishing gears) (*Vasconcelos et al., 2018*), and essential for the proposal of effective measures to protect living resources (e.g., monitoring of stocks and limiting harvesting practices according to size-based analyses of communities) (*Robinson et al., 2010*).

Traditionally, allometric growth analyses are performed using the linear equation and power function (allometric equation) (e.g., *Gaspar, Santos & Vasconcelos, 2001*; *Gaspar et al., 2002*; *MacCord & Amaral, 2005*; *Vasconcelos et al., 2018*). While the former ($y = a + bx$) corresponds to an equivalent increase in size of a body part $y$ (dependent variable) and another body part $x$ (independent variable; the reference dimension), the power function (allometric equation) determines the relationship between two parts of the body through an exponential equation $y = ax^b$, where $b$ is a measure of the constant difference in the growth rates of the body parts $x$ and $y$ (*Katsanevakis et al., 2006*). However, the allometric exponent $b$ is not necessarily constant and may exhibit breakpoints (i.e., points of discontinuity in slope) for instance resulting from marked changes in the environment or physiology (*Katsanevakis et al., 2006*). As an alternative to overcome poor and misleading interpretation of results, more complex models that consider that nonlinearity and breakpoints may exist in the relationship of body parts (e.g., polynomial functions such as quadratic or cubic models) have been recommended (*Hendriks et al., 2012*; *Katsanevakis et al., 2006*). Preferably, researchers should consider a set of pre-established models and use model selection methods, such as the information theory and multi-model inference approach (*Burnham & Anderson, 2002b*), to ascertain which better fits the data (*Katsanevakis et al., 2006*; *Rabaoui et al., 2007*).

Regarding economic important marine species, allometric growth analyses are relatively common for fishes, but much scarcer for benthic invertebrates (*Vasconcelos et al., 2018*). For bivalves, the growth and shape of the shell are normally influenced by both environmental (e.g., temperature, depth, currents, wave exposure, and sediment) and biological factors (predation, growth, and burrowing abilities) (see revision by *Gaspar et al., 2002*). Therefore, variation in these conditions may produce varying growth patterns of the shell and soft parts of the animals. An evaluation of the allometric growth of the bivalve *Pinna nobilis*, for example, registered significant variation in growth among five populations related to

various environmental factors (*Rabaoui et al., 2007*). Marked changes in bivalve allometric growth are also expected to be associated with the maturity of individuals (i.e., the onset of reproduction), given that more energy is allocated to soft parts (mainly to the development and maturation of gonads) than to shell deposition (*Bayne & Worrall, 1980*; *Rabaoui et al., 2007*). To better understand the development of key species and provide critical information for fisheries stock assessment and management, it is therefore essential that allometric growth studies are performed on a wider diversity of organisms and use a more comprehensive set of statistical models. Only through this approach, growth variation among species, and between and within populations (i.e., spatial and temporal comparisons) can be unveiled.

In this study, we tested whether different models are equally adequate to estimate the allometric growth of the trigonal clam *Tivela mactroides* (Born, 1778), an important fishing resource in the Southeastern American coast (*Denadai, Amaral & Turra, 2005*; *Turra et al., 2016*). Specifically, we compared the suitability of models traditionally used in allometric growth analyses (i.e., linear and power function models) with more complex models (i.e., polynomial functions), and investigated whether alternative models may lead to a better interpretation of the data and provide additional biological information about this species.

## MATERIALS AND METHODS

### Model species

The trigonal clam *T. mactroides* is a widespread species found along the South America coast, from Colombia to southeastern Brazil (*Turra et al., 2014*). This bivalve occurs from the shallow subtidal to the upper intertidal zone (*Denadai, Amaral & Turra, 2005*), and is a main feeding resource for several species of fishes, sea-stars, and crabs (*Turra et al., 2015a*). Moreover, *T. mactroides* is economically important and intensively exploited by fishermen and recreational harvesters in many parts of South America (*Turra et al., 2016*).

### Study area

This work was performed at Caraguatatuba Bay, which is located in southeastern Brazil (Fig. 1) and comprises several sandy beaches along a 16 km beach arch (*Denadai et al., 2013*). Wave energy is moderate at Caraguatatuba Bay because of the shadowing effect of São Sebastião Island (*Denadai, Amaral & Turra, 2005*), but beach characteristics vary in a north–south orientation. The southernmost part has a wide intertidal ultradissipative terrace (800 m) with well-sorted fine sand. The northern part has a more heterogeneous slope with low tide dissipative terrace and fine poorly sorted sand (*Denadai, Amaral & Turra, 2005*; *Turra et al., 2014*). The mean seawater temperature in Caraguatatuba Bay ranges from 19 °C in winter to 26 °C in summer (*Corte, 2015*). Salinity is usually above 30 and significant variations only occur near the rivers that flow into the bay (*Amaral & Nallin, 2011*).

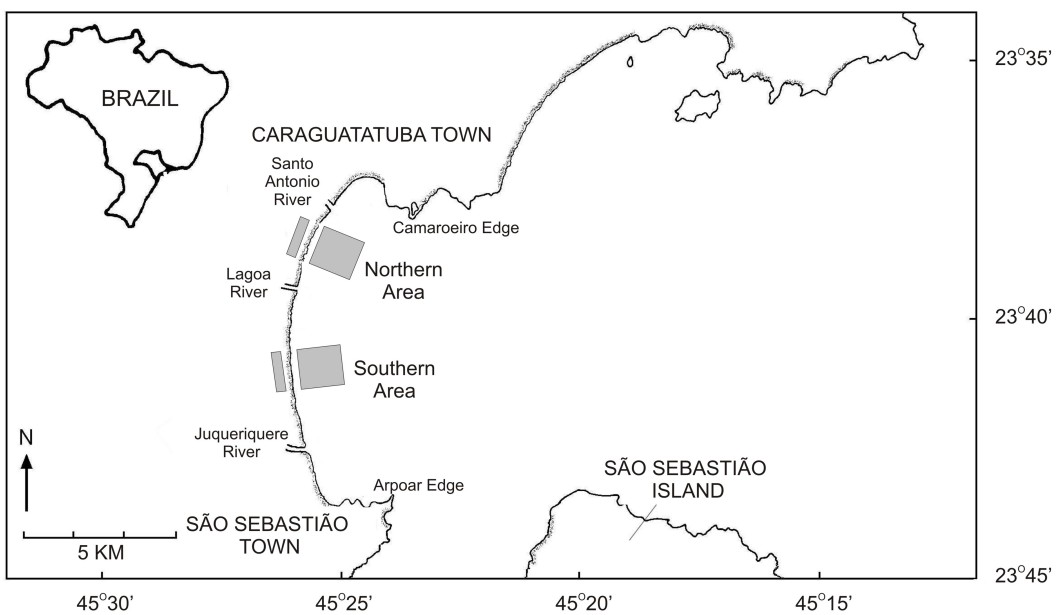

**Figure 1** **Map of the Caraguatatuba Bay, southeastern Brazil.** Sampling areas are highlighted in grey.

## Sampling and laboratory procedures

Field work was done following procedures described in *Turra et al. (2015a)*; *Turra et al. (2015b)*. Briefly, clams were sampled monthly from January 2003 until October 2004 in two areas with high abundance of *T. mactroides* (*Denadai, Amaral & Turra, 2005*) (Fig. 1). The first southern area extended from Porto Novo to Palmeiras; and the northern area from Indaiá to Centro beach (Fig. 1). Both areas had 2,000 m length and sampling was performed in the intertidal and subtidal zone of each area following different procedures.

In the intertidal zone of each area, sampling was performed at eight randomly sorted transects from 200 possibilities (i.e., the linear 2,000 m length divided into 10-m intervals). Six or seven samples (squares of 0.5 × 0.5 m) were excavated to a depth of 10 cm at regular intervals in each transect. The number of samples collected at each transect depended on their length. A total of 120 samples (30 m$^2$) were collected per month in the intertidal area.

In the subtidal zone, sampling was performed using a fishing boat at five different depths, i.e., 400, 800, 1,200, 1,600 and 2,000 m, from mean low water (MLW, 0.0 m). At each depth, one 50-m dredging was performed at three randomly sorted transects perpendicular to the coast. Thus, 30 samples were dredged per month (2 areas × 5 depths × 3 transects). Samples were taken using a rectangular dredge (70 × 25 cm) with 3.0 mm internal mesh size.

The sediment sampled at both intertidal and subtidal zones was washed with seawater over a 3.0-mm-mesh sieve, using buckets. Clams from both zones were collected, counted and measured for shell length to the nearest 0.01 mm with a digital caliper to examine the population dynamics of *T. mactroides* at the study area as reported in *Turra et al. (2014)*. Almost all the individuals were returned alive to the sea after the measurements;

## Tivela mactroides

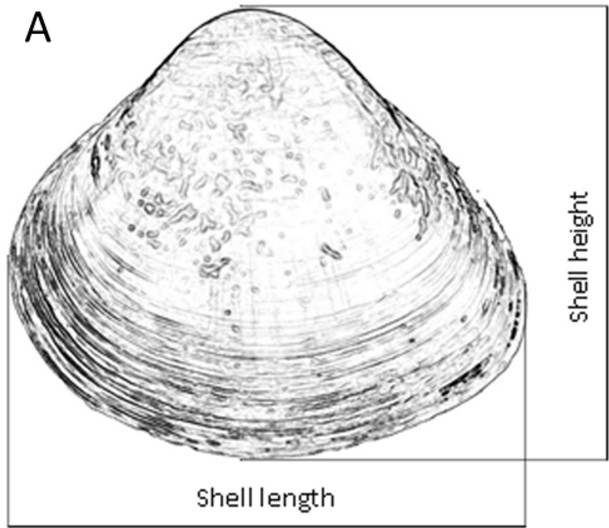
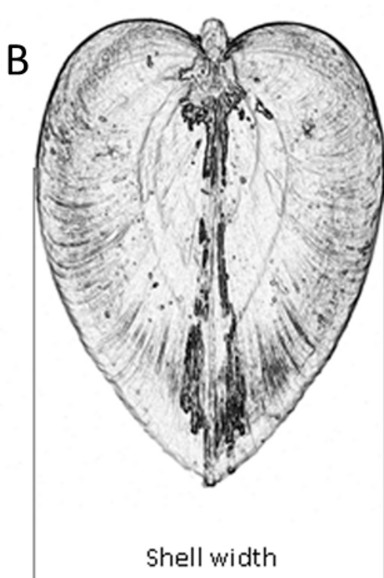

**Figure 2** *Tivela mactroides.* Scheme indicating the measurements taken from the shells (shell length, L; height, H; and width, W): (A) frontal view; (B) lateral view.

however, approximately five individuals from each of the 37 size classes identified in this population (from 3.0 to 39.0 mm; 1 mm intervals) where taken for morphometric analyses, totaling 187 individuals. These specimens were maintained in clean, aerated seawater for 24 h to eliminate the feces. After this period, the soft parts were separated from the shells and dried at 60 °C to achieve a constant weight and afterwards record the shell weight and the soft parts dry weight. Dry weight was used to avoid discrepancy in the amount of water retained or fluctuation in soft-parts weight due to changes in the physiological processes (*Vasconcelos et al., 2008*; *Vasconcelos et al., 2018*). Ash weight was obtained after incinerating the dried soft parts in a muffle furnace at 550 °C for 5 h. The ash-free dry weight (organic content) was calculated by subtracting the ash weight from the dry weight. Thus, all individuals had the shell length ($L$), height ($H$), and width ($W$) measured (Fig. 2), and the shell weight ($SW$), soft parts dry weight ($DW$), and ash-free dry weight ($AFDW$) recorded.

### Data analysis

The allometric growth of *T. mactroides* was analyzed using models that relate the shell length (independent variable) to the shell height, shell width, shell weight, soft parts dry weight, and ash-free dry weight ($L$ vs $H$, $W$, $SW$, $DW$ and $AFDW$). The relationships between shell width (independent variable) and shell height ($W$ vs $H$), and between the soft parts dry weight (independent variable) and the shell weight and ash-free dry weight were also evaluated ($DW$ vs $SW$ and $AFDW$).

We used five models for the comparisons (Table 1): (1) simple linear regression ($y = bx$), (2) second-order polynomial function ($y = bx + cx^2$), (3) third-order polynomial function
**Table 1  Models and their respective equations and number of parameters estimated plus one ($k$).** All models consider origin $= 0$.

| Model | Equation | k |
|---|---|---|
| Simple regression ($Ln$) | $y = bx$ | 2 |
| Second-order polynomial ($Qd$) | $y = bx + cx^2$ | 3 |
| Third-order polynomial ($Cb$) | $y = bx + cx^2 + dx^3$ | 4 |
| Fourth-order polynomial ($Qt$) | $y = bx + cx^2 + dx^3 + ex^4$ | 5 |
| Power-function ($Pf$) | $y = ax^b$ | 3 |

($y = bx + cx^2 + dx^3$), (4) fourth-order polynomial function ($y = bx + cx^2 + dx^3 + ex^4$), and (5) power function (allometric equation, $y = ax^b$). The candidate models for allometric growth were fitted to raw data and the fitted equations were compared based on the Kullback–Leibler (K-L) information theory and multi-model inference (MMI) (*Burnham & Anderson, 2002a*; *Burnham & Anderson, 2002b*). The Kullback–Leibler (K-L) information theory can be interpreted as the distance from the approximating model to full reality, and minimization of K-L distance is essential for model selection (*Katsanevakis et al., 2006*). Multi-model inference, in turn, is a procedure where parameters are estimated from several different models rather than from one *a priori* selected model (*Burnham & Anderson, 2002a*; *Burnham & Anderson, 2002b*).

We used the small-sample bias-corrected form of the Akaike Information Criteria, $AIC_c$ (*Hurvich & Tsai, 1989*) of the $AIC$ (*Burnham & Anderson, 2002a*) for the model selection, according to the equation:

$$AIC_c = AIC + \frac{2k(k+1)}{n - k - 1}$$

where $AIC$ is given by:

$$AIC = n\left(\log\left(2\pi \frac{RSS}{n}\right) + 1\right) + 2k$$

where $RSS$ is the residual sum of squares of the regression, $n$ is the number of observations, and $k$ is the number of estimated parameters of the regression plus 1. The model with the smallest $AIC_c$ value ($AIC_{c,min}$) was selected as the 'best' among the tested models. The $AIC_c$ differences between adjusted functions, $\Delta_i = AIC_{c,i} - AIC_{c,min}$, were computed over all pairs of candidate models. According to *Burnham & Anderson (2002a)* and *Burnham & Anderson (2002b)*, models with $\Delta_i < 2$ have substantial support, models with $4 < \Delta_i < 7$ have considerably less support, and models with $\Delta_i > 10$ essentially have no support and can be ignored. In the present work, we considered that all values of $\Delta_i < 2$ have substantial support for the model, and in the case of multiple models having $\Delta_i < 2$ we opted for the simplest model (principle of parsimony), i.e., the model with fewer estimated parameters.

## Field study permissions

All activities complied with the license issued by the appropriate federal environmental agency (Ministério do Meio Ambiente (MMA)–Instituto Chico Mendes de Conservação da Biodiversidade (ICMBio) No. 19887-1; acronyms for, in English: Ministry of the Environment–Chico Mendes Biodiversity Conservation Institute).

## RESULTS

High determination coeficients ($r^2 > 0.842$) were obtained in all models fitted to the relationships between the morphometric parameters of *T. mactroides* (Table 2). For most relationships, more than one model was suitable since they had values of $\Delta i \leq 2$ (Table 2). The third-order polynomial suitably described all morphometric relationships excluding the *L/W* relationship. Nevertheless, considering the parsimony principle, the traditional power function was overall the most adequate model and better described five morphometric relationships (*L/SW, L/DW, L/AFDW, SW/DW,* and *DW/AFDW*). The third-order polynomial model was the best descriptor for two relationships (*L/H* and *W/H*), whereas the fourth-order polynomial was selected as the best descriptor of the relationships *L/W*.

The relative growth between shell length and height was approximately isometric up to 20 mm L; above this size, shell length increased faster compared to shell height (Fig. 3). The relative growth between shell length and width was always positive allometric (i.e., length increased faster), but this relationship was stronger after reaching maturity (approximately 20 mm in L, (*Prieto, 1980*). The relationships between shell length and soft parts showed that the increase in soft parts (*DW* and *AFDW*) was slow in young individuals (below 20 mm L); followed by a rapid increase in soft parts weight was observed after reaching maturity. A similar relationship was observed between shell length and shell weight. The allometric growth between width and height oscillated during growth, mainly between 15 and 25 mm in shell width. The shell weight increased continuously (practically linearly) with soft parts dry weight. Similar relationship was established between ash-free dry weight and soft parts dry weight.

## DISCUSSION

The usual approach when studying allometric growth in marine species is to *a priori* adopt the classical linear or allometric model (power function), which may have implications in the accuracy and precision of the estimated parameters (*Katsanevakis et al., 2006*; *Rabaoui et al., 2007*). When only these classical models are used, less informative conclusions may be reached by smoothing the real growth pattern (*Protopapas, Thessalou-Legaki & Verriopoulos, 2007*; *Rabaoui et al., 2007*). In this study, we used the information theory and multi-model inference approach (*Burnham & Anderson, 2002a*) to compare the suitability of classical (i.e., linear and power function models) and more complex models (i.e., polynomial functions) to describe the allometric growth of *T. mactroides*. Our results showed that classical models may adequately describe most morphometric relationships of this species. However, their use alone may hide subtle changes in the allometric growth and prevent a full understanding of growth patterns.

The power function (allometric equation) was the best model describing five of the eight morphometric relationships analyzed (mainly relationships involving the weight parameters such as shell weight, dry weight and ash-free dry weight). This function is commonly used in morphometric analyses of bivalves (e.g., *Clasing et al., 1994*; *Urban & Campos, 1994*; *Gaspar, Santos & Vasconcelos, 2001*; *Lomovasky, Brey & Morriconi, 2005*),

**Table 2** *Tivela mactroides.* Summary and comparison among the determination coefficient ($r^2$), standard error of the estimate (SE), residual sum of squares (RSS), Akaike information criteria corrected for small samples ($AIC_c$), and difference between $AICc$ ($\Delta i$) of all models.

| Relation | Model | $r^2$ | SE | RSS | AICc | $\Delta i$ |
|---|---|---|---|---|---|---|
| L/W | Ln | 0.997 | 0.857 | 136.871 | 476.392 | 94.070 |
| | Qd | 0.998 | 0.753 | 104.985 | 428.861 | 46.539 |
| | Cub | 0.998 | 0.689 | 87.519 | 396.923 | 14.602 |
| | **Qt** | **0.998** | **0.661** | **80.037** | **382.322** | **0.000** |
| | Pf | 0.985 | 0.799 | 118.223 | 451.068 | 68.747 |
| L/H | Ln | 0.999 | 0.730 | 99.106 | 416.019 | 107.264 |
| | Qd | 0.999 | 0.574 | 61.111 | 327.669 | 18.914 |
| | **Cub** | **0.999** | **0.546** | **55.005** | **310.073** | **1.318** |
| | Qt | 0.999 | 0.543 | 54.005 | 308.755 | 0.000 |
| | Pf | 0.996 | 0.617 | 70.424 | 354.194 | 45.439 |
| L/SW | Ln | 0.861 | 1.730 | 556.922 | 738.824 | 464.003 |
| | Qd | 0.987 | 0.530 | 52.115 | 297.891 | 23.070 |
| | Cub | 0.989 | 0.499 | 45.964 | 276.496 | 1.675 |
| | Qt | 0.989 | 0.501 | 45.942 | 278.516 | 3.695 |
| | **Pf** | **0.976** | **0.499** | **46.066** | **274.821** | **0.000** |
| L/DW | Ln | 0.844 | 0.094 | 1.637 | −342.574 | 250.325 |
| | Qd | 0.959 | 0.048 | 0.429 | −586.848 | 6.052 |
| | Cub | 0.960 | 0.047 | 0.415 | −591.088 | 1.811 |
| | Qt | 0.961 | 0.047 | 0.411 | −590.689 | 2.210 |
| | **Pf** | **0.918** | **0.047** | **0.415** | **−592.899** | **0.000** |
| L/AFDW | Ln | 0.842 | 0.086 | 1.351 | −374.856 | 251.689 |
| | Qd | 0.959 | 0.044 | 0.351 | −619.540 | 7.004 |
| | Cub | 0.961 | 0.043 | 0.337 | −624.709 | 1.836 |
| | Qt | 0.961 | 0.043 | 0.334 | −624.112 | 2.433 |
| | **Pf** | **0.918** | **0.043** | **0.338** | **−626.544** | **0.000** |
| W/H | Ln | 0.999 | 0.835 | 62.982 | 331.243 | 20.198 |
| | Qd | 0.999 | 0.837 | 61.612 | 329.198 | 18.153 |
| | **Cub** | **0.999** | **0.830** | **55.291** | **311.045** | **0.000** |
| | Qt | 0.999 | 0.7651 | 107.127 | 436.837 | 125.792 |
| | Pf | 0.992 | 0.8365 | 129.446 | 468.027 | 156.982 |
| DW/SW | Ln | 0.962 | 0.047 | 0.399 | −597.694 | 0.266 |
| | Qd | 0.962 | 0.047 | 0.395 | −597.676 | 0.285 |
| | Cub | 0.963 | 0.046 | 0.393 | −596.381 | 1.580 |
| | Qt | 0.963 | 0.047 | 0.391 | −595.099 | 2.861 |
| | **Pf** | **0.928** | **0.047** | **0.394** | **−597.961** | **0.000** |
| DW/AFDW | Ln | 0.999 | 0.005 | 0.004 | −1432.479 | 5.734 |
| | Qd | 0.999 | 0.005 | 0.004 | −1434.282 | 3.932 |
| | Cub | 0.999 | 0.005 | 0.004 | −1438.214 | 0.000 |

**Table 2** (*continued*)

| Relation | Model | $r^2$ | SE | RSS | AICc | $\Delta i$ |
|---|---|---|---|---|---|---|
| | *Qt* | 0.999 | 0.005 | 0.004 | −1436.168 | 2.045 |
| | **Pf** | **0.999** | **0.005** | **0.004** | **−1437.069** | **1.145** |

**Notes.**

L, shell length; W, shell width; H, shell height; SW, shell weight; DW, soft parts dry weight; AFDW, ash-free dry weight; Ln, linear model; Qd, second-order polynomial model; Cub, third-order polynomial model; Qt, fourth-order polynomial model; Pf, power function.

Most suitable models according to AICc and parsimony principle are highlighted in bold. When $\Delta_i < 2$, the model with the smallest number of parameters was selected.

but is normally considered the weakest adjustment for many morphometric relationships in some marine invertebrates (*Katsanevakis, 2007*; *Rabaoui et al., 2007*; *Garraffoni, Yokoyama & Amaral, 2010*). A main drawback in the use of the power function is that it does not consider breakpoints (i.e., marked changes in the allometric growth) and may hide major indications of morphological variability among individuals of different sizes (*Rabaoui et al., 2007*). Our results show that the power function best characterized the allometric growth between shell parameters and soft tissues, indicating that shell and somatic parts of *T. mactroides* grow with a constant allometric exponent, whereas polynomial models better described relationships between shell parameters. Likewise, *Trussell (2000)* found that changes in shell morphology in *Littorina* snails were not related to soft tissue variations, and *McKinney, Glatt & Williams (2004)* reported that allometric regression models best described changes in soft tissue content with shell length for 10 different species, including the ribbed mussel *Geukensia demisa*, the blue mussel *Mytilus edulis*, and the quahog *Mercenaria mercenaria*.

Several factors may influence the growth of soft tissue relative to shell parameters. Food availability, for instance, can strongly affect growth, storage and utilization of body reserves; thereby altering allometric relationships (*McKinney, Glatt & Williams, 2004*). Similarly, temperature and salinity may alter metabolic rates and determine net growth efficiency in bivalve species (*Resgalla Jr, Brasil & Salomão, 2007*; *Xiao et al., 2014*). The constant allometric growth of shell measurements and weighings of *T. mactroides* at Caraguatatuba Bay is likely related to constant food supply to the clams due to the presence of three rivers (i.e., Santo Antônio, Lagoa and Juqueriquerê—Fig. 1) that flow directly into the bay and provide a large amount of suspended organic material (*Corte, 2015*; *Turra et al., 2015b*). The absence of marked fluctuation in seawater temperature and salinity at the study area (*Amaral & Nallin, 2011*; *Corte, 2015*) probably also contributes to the constant growth of soft tissues.

Although the power function was more precise to describe the morphometric relationship between shell parameters and somatic parts, it was not enough to unveil changes in the morphometric relationships between shell parameters. By adjusting polynomial functions, we were able to detect differences in morphometric relationships of shell parameters over the lifespan of *T. mactroides*. Changes in bivalves morphometric relationships are usually related to the maintenance of an area/volume ratio that is physiologically suitable for the prevailing environmental conditions (*Gaspar et al., 2002*; *Rhoads & Pannella, 1970*). For instance, the shell of the pill clam *Pisidium subtruncatum*

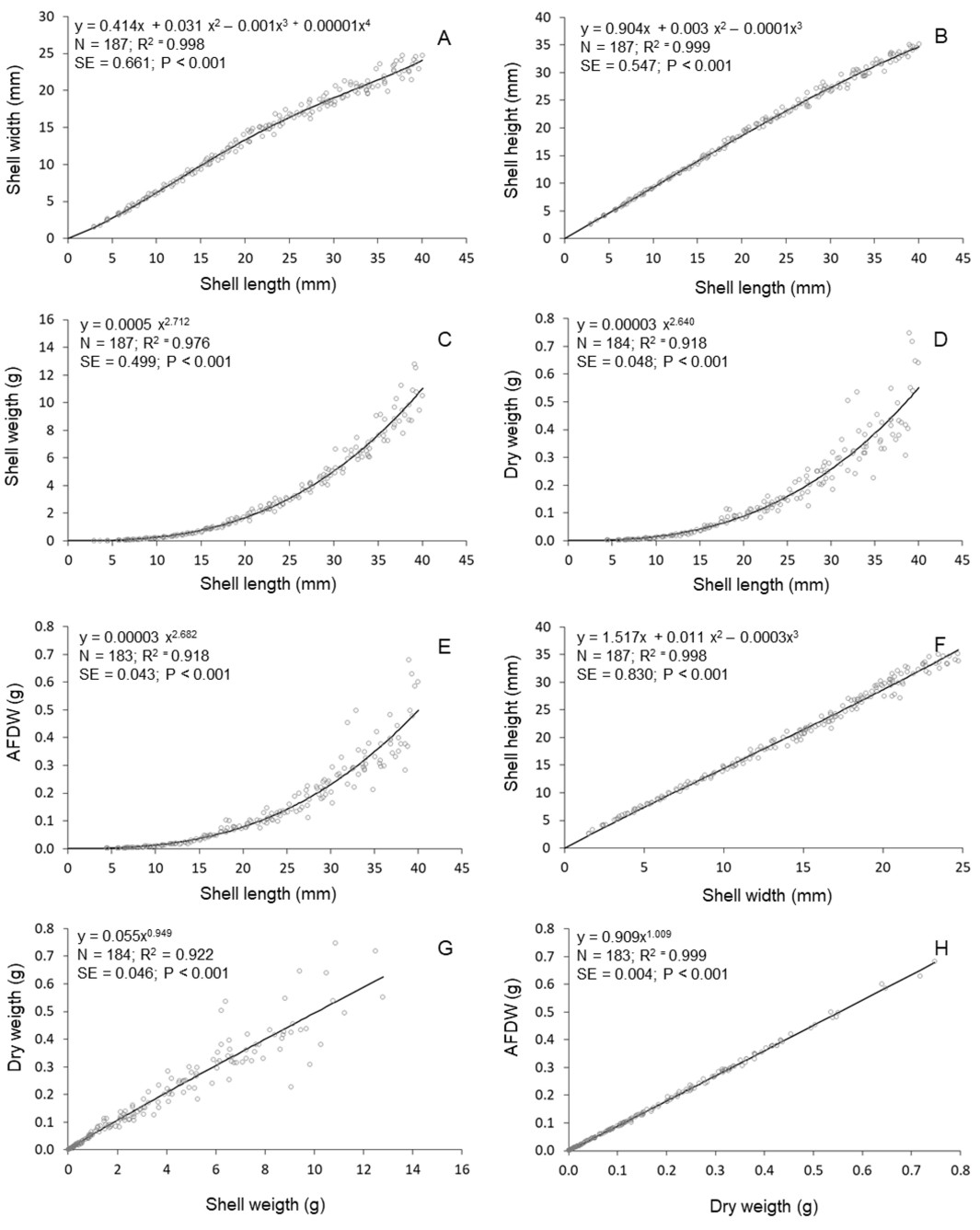

**Figure 3** *Tivela mactroides.* Graphical representations of the allometric relationships. Continuous lines represent the most suitable model describing the morphometric relationship. N, number of individuals analyzed; SE, standard error; AFDW, ash-free dry weight. (A) Shell width ∼ shell length; (B) shell height ∼ shell length; (C) shell weigth ∼ shell length; (D) dry weigth ∼ shell length; (E) ash free dry weigth ∼ shell length; (F) shell heigth ∼ shell width; (G) dry weigth ∼ shell weigth; (H) ash free dry weigth dry weigth.

(Malm, 1855) change from a rounded and thin-shelled form to an angular and thick-shelled form depending on the proportion of sand in the sediment (*Funk & Reckendorfer, 2008*), which enables the individuals to burrow easier in coarse sediments. Similar modifications have also been observed in species that change their habits from active burrowing juveniles to sedentary and deeper burrowing adults (*Statzner & Holm, 1982*), such as the razor clam *Ensis macha* that shows an abrupt change in shell shape during growth, from squarer to more elongated (*Barón et al., 2004*). This elongated form enables razor clams to avoid predators by burrowing deeper with low energy requirements (*Trueman, 1967*; *Urban, 1994*).

Modifications in the allometric relationships between shell parameters of *T. mactroides* seem related to habitat changes and maturity of individuals. We observed that shell width and height had a reduction in their proportional growth in relation to shell length when individuals reached a length between 15 and 25 mm. This change is appropriate for the subtidal-intertidal migration undertaken by this species (*Denadai, Amaral & Turra, 2005*). The smallest individuals are usually found in mud sediments in the subtidal level, where burrowing with a more-inflated shell is easier. As individuals reach maturity, they passively migrate to firm intertidal sandy substrata and their shell becomes longer and more compressed, potentially facilitating burrowing in this type of sediment. Maturity is also a frequent cause of a distinct change in morphology, given that more energy is allocated to the development of gonads than to shell growth (*Bayne & Worrall, 1980*; *Katsanevakis et al., 2006*).

Besides providing further information for understanding the life history of species, the knowledge of allometric growth has implications for economic exploitation of fishing resources. In bivalves, the morphometric relationships between shell size and soft parts can help defining an optimum size for exploitation. For *T. mactroides*, the analyses of allometric growth indicate that more energy is invested in shell growth during early phases of the life cycle, thereby increasing the protection and survival of juveniles. Only after reaching maturity, a faster increase in somatic tissues was recorded. Thus, harvesting of *T. mactroides* should preferably target individuals with shell length between 25–30 mm. This would ensure that individuals are already able to reproduce, thus providing a better income to local harvesters since a higher meat yield (ratio meat weight/shell weight) would be achieved.

## CONCLUSIONS

Our results show that the classical power function is useful to evaluate the relative growth of *Tivela mactroides*. Nevertheless, the use of this model alone may hide subtle changes in morphology related to environmental or physiological processes such as habitat changes and maturity of individuals. Only by applying more complex models, in the present case polynomial models, we perceived subtle modifications in the allometric growth of *T. mactroides*. In this regard, we reinforce the recommendation that allometric growth analyses should consider a set of pre-established models including traditional and more complex models. Certainly, this would improve allometric analyses and provide stronger and more informative conclusions.

## ACKNOWLEDGEMENTS

We are grateful to all the staff who worked on the Biota/Fapesp Bentos-Marinho project (Proc 1998/07090-3).

### Funding

This work was supported by the Fundação de Apoio à Pesquisa do Estado de São Paulo (FAPESP) (Proc. 06/57575-1; 05/60041-6; 2017/17071-9), the Project AWARE Foundation, the PADI Foundation, and the Conselho Nacional de Desenvolvimento Científico e Tecnológico (CNPq) (Proc. 150473/2010-9; 306534/2015-0; 309697/2015-8). The funders had no role in study design, data collection and analysis, decision to publish, or preparation of the manuscript.

### Grant Disclosures

The following grant information was disclosed by the authors:
Fundação de Apoio à Pesquisa do Estado de São Paulo (FAPESP): 06/57575-1, 05/60041-6, 2017/17071-9.
AWARE Foundation.
PADI Foundation.
Conselho Nacional de Desenvolvimento Científico e Tecnológico (CNPq): 150473/2010-9, 306534/2015-0, 309697/2015-8.

### Competing Interests

The authors declare there are no competing interests.

### Author Contributions

- Alexander Turra conceived and designed the experiments, performed the experiments, analyzed the data, contributed reagents/materials/analysis tools, prepared figures and/or tables, authored or reviewed drafts of the paper, approved the final draft.
- Guilherme N. Corte prepared figures and/or tables, authored or reviewed drafts of the paper, approved the final draft.
- Antonia Cecília Z. Amaral conceived and designed the experiments, performed the experiments, contributed reagents/materials/analysis tools, authored or reviewed drafts of the paper, approved the final draft.
- Leonardo Q. Yokoyama analyzed the data, prepared figures and/or tables, authored or reviewed drafts of the paper, approved the final draft.
- Márcia R. Denadai conceived and designed the experiments, performed the experiments, analyzed the data, contributed reagents/materials/analysis tools, authored or reviewed drafts of the paper, approved the final draft.

### Field Study Permissions

The following information was supplied relating to field study approvals (i.e., approving body and any reference numbers):

All activities complied with the license from the appropriate federal environmental agency (Ministério do Meio Ambiente (MMA)—Instituto Chico Mendes de Conservacão da Biodiversidade (ICMBio); acronyms for, in English: Ministry of the Environment—Chico Mendes Biodiversity Conservation Institute) (No. 19887-1).

## Data Availability

All measurements from this work are provided as Data S1.

## Supplemental Information

Supplemental information for this article can be found online at http://dx.doi.org/10.7717/peerj.5070#supplemental-information.

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
