# Peer review of "Non-linear curve adjustments widen biological interpretation of relative growth analyses of the clam Tivela mactroides (Bivalvia, Veneridae)"

_PeerJ, doi:10.7717/peerj.5070_

## Round 0.1 · original submission · Major Revisions

· Academic Editor

Major Revisions

I agree with all reviewers when they state that your MS provide interesting and valuable information regarding the use of different models to assess the allometric growth of bivalves and specifically for Tivela mactroides. Also, the problem is correctly presented for the results that are shown.

However, I also agree when, especially Reviewer 1 and 2, request for more information regarding the sampling points and the samples numbers, which means that you should provide more information on the sampling design and methods used to collect the specimens, as well as the reasons behind the limited number of data used to produce the analysis. The discussion should be revised in order to better explain and compare the differences between the classic and the more complex models.

I therefore invite you to consider their comments and prepare a revised version of the manuscript that you submit, along with a rebuttal of how you have addressed their concerns. Thus, depending on your rebuttal and the revisions made I will consider if a second revision is necessary.

Reviewer 1 ·

Basic reporting

No comment.

Experimental design

No comment.

Validity of the findings

No comment.

Additional comments

MS Ref: PeerJ_24725
Title: “Non-linear curve adjustments widen biological interpretation of relative growth analysis of the harvested clam Tivela mactroides (Bivalvia, Veneridae)"
Authors: Alexander Turra, Guilherme N. Corte, A. Cecília Z. Amaral, Leonardo Q. Yokoyama, Márcia R. Denadai


REVIEWER COMMENTS:

The present manuscript submitted to “PeerJ” compares different types of models (simple linear, power and polynomial models) to assess allometric growth of Tivela mactroides, aiming to improve the biological information gathered in studies of relative growth.

Overall, the manuscript is well written, organised and presented, providing interesting and valuable information on this research topic. Accordingly, I support the acceptance of the manuscript for publication in “PeerJ” following major revision (please see the general and specific comments listed below).

General comments:

Since the trigonal clam (T. mactroides) is a locally important fishing resource along the Brazilian coast, targeted by fishermen and recreational harvesters, the number of specimens analysed in the study is surprisingly low. In fact, after one-year of monthly sampling (Nov 2003 - Oct 2004) only 187 individuals were sampled (≈ 15 individuals per month). Nevertheless, the samples comprised individuals with a broad size range (3.0–39.0 mm in shell length) and the adjustment of the models presented invariably high determination coefficients and statistical significance levels. Still, I invite the authors to explain why such a “limited” data-set was used in the present study.

The results revealed that “the power function was more suitable to describe the relationships involving shell length and weight parameters”, i.e. the “classic” model (Y = aXb) for analysing relative growth remained the best approach for assessing isometry vs. allometry in relationships involving both linear and ponderal variables. On the opposite, “morphometric relationships between shell parameters… were better represented by polynomial functions”. In this context, I invite the authors to improve their discussion on this subject, by explaining why relationships between shell measurements (linear variables) required more complex models (polynomial functions), whereas in relationships between shell measurements (linear variables) and weighings (ponderal variables), the “classic” power function was the most suitable model and more complex models did not provide any added-value.

In the Discussion section, authors limit their explanations to the results obtained in the present study and do not provide comparisons with analogous studies performed with this species. For this reason, if available, I invite the authors to add and compare data gathered in previous studies on morphometric / allometric relationships in T. mactroides. Ideally, authors should highlight eventual differences resulting from the adjustment of different models (“classic” vs. “complex”) and the main advantages of adopting the present approach in terms of the information obtained about morphometric trends during growth (i.e. ontogenetic shifts) related to changes in the species biology / ecology.

Specific comments:

Title:
- I suggest removing the word “harvested” from the title, because it does not add relevant information about the objective and main conclusions of the study.

Abstract:
- line 18: Please replace “economically exploited” by “fishery-exploited”.
- line 20: Please replace “Nevertheless” by “However”.
- line 22: Please replace “overcome poor and misleading results” by “overcome poor results and misleading interpretations”.
- line 26: Please replace “fishery resource” by “fishing resource”.
- lines 27-28: Please replace “to each other and to soft parts (dry weight (DW)” by “among each other and with soft parts of the organism (dry weight (DW)”.
- line 31 (and throughout the entire text): Please replace “(i.e.,) by “(i.e.).
- line 33: Please replace “represented” by “described”.
- lines 35-36: Please replace “benthic invertebrates” by “marine bivalves”.
- line 38: Please replace “benthic invertebrates” by “molluscs”.

Introduction:
- line 44: Please replace “permits” by “allows”.
- line 47: Please replace “allows” by “provides”.
- line 50 (and throughout the entire text): Please replace “(e.g.,) by “(e.g.).
- line 50: Please replace “shortage of food” by “food availability”.
- line 50: Please replace “spawning periods” by “gonad maturation and spawning periods”.
- line 52: Please replace “in the ontogeny” by “during the ontogeny”.
- line 54: Please replace “selectivity of dredges and meshes” by “size-selectivity of fishing gears”.
- line 59: Please replace “Gaspar et al. 2001; Gaspar et al. 2002;” by “Gaspar et al. 2001, 2002;”.
- line 61: Please replace “another part x” by “another body part x”.
- line 64: Please replace “the two body parts” by “the body parts x and y”.
- line 65: Please replace “Nevertheless” by “However”.
- lines 66-67: Please replace “resulting from marked changes in the environment or physiology, for example” by “for instance resulting from marked changes in the environment or physiology”.
- line 73: Please replace “to find out which one better fits” by “to ascertain which better fits”.
- line 79: Please replace “Therefore, the variation” by “Therefore, variation”.
- line 81-82: Please replace: “growth of five different populations of the bivalve Pinna nobilis, for example, registered significant variation in the growth among the populations” by “growth of the bivalve Pinna nobilis for example, registered significant variation in growth among five populations”.
- line 85: Please replace “development of gonads” by “development and maturation of gonads”.
- line 86: Please replace “than to the shell” by “than to shell deposition”.
- line 88: Please replace “are done with a large” by “are performed on a wider”.
- line 93: Please replace “Tivela mactroides,” by “Tivela mactroides (Born, 1778),”.
- line 93: Please replace “fishery resource” by “fishing resource”.
- line 94: Maybe “eastern South American coast” is better described as “Southeastern America coast”. Please clarify.
- lines 96-97: Please replace “investigated if alternate models” by “investigated whether alternative models”.
- line 97: Please replace “more biological” by “additional biological”.

Materials and Methods:
- line 102: Please replace “Tivela mactroides” by “T. mactroides”.
- line 104: Please replace “food resource” by “feeding source”.
- line 106: Please replace “tourists” by “recreational harvesters”.
- line 110: Please replace “done” by “performed”.
- line 111: Please replace “contains” by “comprises”.
- line 117: Please replace “the dry weight of the soft parts” by “the soft parts dry weight”.
- lines 119-120: Please replace “(Vasconcelos et al. 2008; Vasconcelos et al. 2016)” by “(Vasconcelos et al. 2008, 2016).”
- lines 123-124: Please replace “dry weight of the soft parts” by “soft parts dry weight”.
- line 124: Please replace “calculated” by “recorded”.
- line 127: Please replace “that related” by “that relate”.
- lines 128, 130, 131: Please replace “(reference or independent variable)” by “(independent variable)”.
- lines 128-129: Please replace “dry weight of the soft parts” by “soft parts dry weight”.
- line 131: Please replace “the dry weight of the soft parts” by “the soft parts dry weight”.
- line 137: Please replace “fitted curves” by “fitted equations”.
- line 151: Please replace “case of more than one having” by “multiple models having”.
- lines 152-153: Please replace “which was the one with the smallest number of parameters to be estimated.” by “i.e. the model with fewer estimated parameters”.
- line 156: Please replace “from the” by “issued by”.

Results:
- line 162: Please replace “High values of r2 (>0.842) were found in all the models” by “High determination coefficients (r2 > 0.842) were obtained in all models”.
- line 172: Please replace “20 mm; after this size,” by “20 mm L; above this size,”.
- line 172: Please replace “faster in relation to the shell height” by “faster compared to shell height”.
- line 174: Please replace “after maturity” by “after reaching maturity”.
- line 176: Please replace “(with shell length less than 20 mm)” by “(below 20 mm L)”.
- line 177: Please replace “in weight of soft parts” by “in soft parts weight”.
- line 177: Please replace “after maturity” by “after reaching maturity”.
- line 179: Please replace “during the development of individuals” by “during growth”.
- line 180: Please replace “dry weight of soft parts” by “soft parts dry weight”.
- line 181: Please replace “in relation to the dry weight of soft parts” by “in relation to soft parts dry weight”.

Discussion:
- line 189: Please replace “true pattern” by “real growth pattern”.
- line 192: Please replace “growth of the trigonal clam T. mactroides.” by “growth of T. mactroides.”.
- line 195: Please replace “prevents” by “prevent”.
- line 198: Please replace “as dry weight, shell weight, and” by “as shell weight, dry weight and”.
- line 201: Please replace “for many of the morphometric relationships in marine invertebrates” by “for many morphometric relationships in some marine invertebrates”.
- line 204: Please replace “among the individuals” by “among individuals”.
- lines 205-206: Please replace “between soft tissues and shell parameters” by “between shell parameters and soft tissues”.
- lines 208-210: Please replace “inhabited by T. mactroides. This species occurs in relatively exposed sandy beaches with an abundant quantity of suspended material (Narchi 1972), which may supply the
species with enough food” by “inhabited by T. mactroides, which occurs in relatively exposed sandy beaches with abundant suspended material (Narchi 1972) that supplies enough food”.
- line 211: Please replace “constant supply of food” by “constant food supply”.
- line 214: Please replace “consequence of living in disturbed waters” by “consequence of inhabiting disturbed waters”.
- line 219: Please replace “By using polynomial functions” by “By adjusting polynomial functions”.
- line 223: Maybe by “migration” authors mention a depth segregation between juveniles in the subtidal and adults in the intertidal. In this case, maybe authors could replace “migration” by “subtidal-intertidal migration throughout ontogeny”. Please clarify.
- line 225: Please replace “tide level” by “tidal level”.
- line 226: Please replace “(burrowing behavior, for example)” by “(e.g. burrowing behaviour)”.
- line 227: Please replace “Alterations in the morphometric relationships of bivalves” by “Alterations in bivalves morphometric relationships”.
- line 228: Please replace “the environmental conditions” by “the prevailing environmental conditions”.
- lines 229-230: Please replace “The shell of the pill clam Pisidium subtruncatum (Malm 1855), for example, change” by “For instance, the shell of the pill clam Pisidium subtruncatum (Malm 1855) changes”.
- lines 231-232: Please replace “This change enables the individuals to burrow in coarse sediments easier.” By “, which enables the individuals to burrow easier in coarse sediments.”
- line 233: Please replace “species which change” by “species that change”.
- line 235: Please replace “abrupt change in the shell shape,” by “abrupt change in shell shape during growth,”.
- lines 235-236: Please replace “from squarer to more elongated, as the individuals grow (Barón et al. 2004).” by “from squarer to more elongated (Barón et al. 2004).”
- line 238: Please see the comment provided above regarding the “migration”. Maybe authors could replace “migration process done by this species” by “subtidal-intertidal migration undertaken by this species”. Please clarify.
- lines 242-243: Please replace “Maturity of individuals is also a” by “Maturity is a”.
- line 244: Please replace “than to the shell” by “than to shell growth”.
- lines 246-247: Please replace “in their allometric growth during their life cycle” by “in allometric growth during their life cycles”.
- line 247: Please replace “utmost importance to be able to detect” by “utmost importance to detect”.
- line 248: Please replace “adequate represent” by “adequately represent”.
- line 250: Please replace “to locate the discontinuities” by “to discern discontinuities”.
- line 255: Please replace “bringing important knowledge for the understanding of the life history” by “bringing further information for understanding the life history”.
- line 256: Please replace “living resources” by “fishing resources”.
- line 259: Please replace “show that more” by “indicate that more”.
- line 260: Please replace “maximizes its shell growth” by “maximizes shell deposition”.
- line 262: Please replace “Only after maturity, a faster increase in somatic elements” by “Only after reaching maturity, a faster increase in somatic tissues”.
- line 263: Please replace “should include individuals with shell length between 25-30 mm length preferably.” by “should target individuals with shell length between 25-30 mm length.”.
- line 264: Please replace “individuals were able” by “individuals are already able”.
- line 265: Please replace “local clammers since a better ratio meat/shell would be achieved.” by “local harvesters since a higher meat yield (ratio meat weight/shell weight) would be achieved.”

Conclusions:
- line 269: Please replace “of the harvested clam Tivela mactroides” by “of T. mactroides”.
- line 270: Please replace “environmental or physiological process.” by “environmental or physiological processes.”.
- line 271-72: Please replace “we perceived modifications” by “we perceived subtle modifications”.
- line 275: Please replace “more informative results” by “more informative conclusions”.

Tables:
- Tables 1 and 2: Please standardise the notation / abbreviation used for the third-order polynomial equation, since it is presented as “(Cb)” in Table 1 and as “(Cub)” in Table 2. Please revise and correct accordingly.
- Table 2 - line 1: Please replace “coefficient of determination” by “determination coefficient”.
- Table 2 - line 3: Please replace “of all the models” by “of all models”.
- Table 2 - line 4: Please replace “DW, dry weight of the soft parts” by “DW, soft parts dry weight”.
- Table 2 - line 5: Please revise whether it should be “(Cb)” as Table 1 or “(Cub)” as in Table 2.
- Table 2 - line 7: Please replace “are in bold” by “are highlighted in bold”.
- Table 2 - Column “SE”: Whenever possible, please use only 3 decimal places in the SE values (just like in all other columns presented in Table 2).

Figures:
- Figure 1 - Map: Maybe authors could improve the map by indicating the main areas where the specimens were collected.
- Figure 2 - Legend: Please replace: “Scheme indicating the measurements” by “Schematic representation of the measurements”.
- Figure 3 - Legend: Four graphs display b values (“b>1*” and “b<3***”) to highlight the type of relative growth between variables (i.e. “b>1” = positive allometry and “b<3” = negative allometry) in the models adjusted using the power function (Y = aXb). In addition, I presume that the asterisks denote the statistical significance (* = p < 0.05 and *** = p < 0.001) of those allometry coeeficients. This information (b values and respective asterisks) must be added in the legend of Figure 3, duly described in the Results section and explained in the Discussion section.
- Figure 3 - Models equations and SE values: Whenever possible, please use only 3 decimal places (just like in the r2 values and p values displayed in Figure 3 and in almost all columns presented in Table 2).
- Figure 3 - Graphs Y-axes: In the five graphs with values above 1, please remove the unnecessary decimal places.
- Figure 3 - Graph DW vs. SW: In order to keep conformity with all the other graphs, I suggest changing the order of the variables (i.e. X = shell weight; Y = dry weight) and accordingly revise the parameters of the adjusted model.

Annotated reviews are not available for download in order to protect the identity of reviewers who chose to remain anonymous.

Reviewer 2 ·

Basic reporting

This article reports the findings of an assessment of non-linear curve adjustments of relative growth analyses of Tivela mactroide. The paper is interesting and revealed the importance that classical and more complex models in studies of allometric growth of benthic invertebrates.

The article is clearly laid and all the key elements are present (abstract, introduction, methodology, results and discussion).

Please carefully read through the manuscript and correct inconsistencies and issues with orthography - grammar and style improvement (especially space between words). Overall, the paper is writen in clear English, mistakes are comparatively rare.

Experimental design

This is a original primary research within aims and scope of the PeerJ. The research question well defined, relevant and meaningful. However, some questions about methods and experimental design need to be clearer.

The sampling design and sampling area were not described and contain no information about replicates. It is not possible to understand the sites where the sampling was made neither by the map or by the succinct description of the study area.
The authors mentioned in the introduction that the growth and shape of the shell are normally influenced by environmental factors, but there is nothing about characterization of the physical variables in the study area. Although the physical variables characterization was not one of the aims, it is important to characterize the environment to strengthen the hypotheses addressed in the discussion.

The number of individuals was extremely low for allometric growth analysis. It was used approximately 15 individuals per month and only 5 individuals per size class. In one of the cited studies (Gaspar et al 2002), the shell morphometric was performed on 25 species with a measured N of approximately 300 individuals per species.

Validity of the findings

The authors used the small-sample bias-corrected form of the Akaike Information Criteria, AICc (Hurvich & Tsai 1989) of the AIC (Burnham & Anderson 2002a) for the model selection, but why not to use Akaike (AICw)? Weights measure the relative likelihood of a model being the best for the given data. This is a straight-forward procedure in a multimodel inference routine (Burnham & Anderson, 2002).

The discussion, in general, is suggestive and repetitive. The discussion still needs some work on the flow of logic and clarity of ideas. Example, in first paragraph Ln 199-202: “The usual approach when studying allometric growth in marine species is to a priori adopt the classical linear or allometric model (power function), which may have implications in the accuracy and precision of the estimated parameters (Katsanevakis et al. 2006; Rabaoui et al. 2007)”. The authors can not state this based on their own results. Depends on the selected parameters, in its study the power function was more suitable to describe all the relationships involving shell length and weight parameters.

Additional comments

Maturity of individuals is also a frequent cause of a distinct change in morphology, given that more energy is allocated to the development of gonads than to the shell (besides considerable changes in weight). Would not it be more interesting to consider sex in the analysis?

The authors mentioned that allometric growth analyses are performed using the linear equation and power function; and that these equations do not consider discontinuities in growth, besides may mask important biological information, however, it was the best model selected for half of the relations evaluated, especially relationships involving shell length and weight parameters. What do you suggest about it?

Time is really an important factor in growth analysis. Do you consider a year as the ideal period for this kind of study? In addition to the allometric growth, individuals were separated by size class for the one of the novelties of the study: evaluate relationships between the dry weight of soft parts and the shell weight and ash-free dry weight. Is an N of 5 individuals sufficient to characterize the population standard?

Annotated reviews are not available for download in order to protect the identity of reviewers who chose to remain anonymous.

·

Basic reporting

The manuscript is well-written; language is concise, unambiguous and technically correct. It also conforms to technical standards of courtesy and expression.

The manuscript includes a sufficient literature review to provide adequate introduction and background relative to recent approaches in the study of allometric growth in bivalves. Relevant literature seems appropriate referenced; however, the titles of the articles in the list of references appear either as sentences (e.g. Corte 2015) or as titles (i.e., capital as the first letter of each word, see Caill-Milly et al. 2014).

The three figures are relevant to the content of the article, they are appropriately described and labeled and their quality and resolution are good though could be improved; for example, equations in Fig. 3 would be clearer using a larger font and fewer decimals. Both tables are also relevant and clear. The provided raw data file is complete and clear.

I believe that the article is self-contained in the sense that all results relevant to its hypothesis are included.

Experimental design

The research question, i.e. the potential shortcomings of “traditional” allometric growth models, is well within the scope of PeerJ. The topic is well defined, relevant and meaningful, and the results of this case of study are valuable. The investigation was conducted rigorously, and the methods are described with sufficient information and clarity to be replicated.

Validity of the findings

Data used in this article are statistically robust; the number of examined organisms is large, and the size-classes are well represented within the sample. The methods and criteria used for selecting the best model in each case are also adequate. In particular, it is commendable that the principle of parsimony was applied even though the alternative models that the authors are proposing are the less parsimonious. Even though, the authors were able to identify minor, subtle changes in growth which could be meaningful, and which are not detectable by the traditional approach. Up to this point the conclusions are properly stated and supported.

Additional comments

This submission is a good example that the scope and decision criteria of PeerJ are adequate. For the most part the results were negative in the sense that the “traditional” power function model showed the best fit to the data. If negative results were not accepted, this article could be rejected based on a subjective determination of low impact or not enough novelty. Though the conclusions are properly stated and supported, and since PeerJ welcomes speculation as long as it is clearly identified as such, I would have liked to see a more extensive discussion on the meaning and possible causes of the subtle changes in allometric growth.

---

## Round 0.2 · Minor Revisions

· Academic Editor

Minor Revisions

I'm pleased with the revised version and the way the reviewers comments were addressed in the revised manuscript and feel that your manuscript makes a useful contribution to this field of research. There are only some minor suggestions made by one of the reviewers which should be attended to before we can Accept the manuscript

Reviewer 1 ·

Basic reporting

Please see "General comments for the authors"

Experimental design

Please see "General comments for the authors"

Validity of the findings

Please see "General comments for the authors"

Additional comments

MS Ref: PeerJ_24725_v1
Title: “Non-linear curve adjustments widen biological interpretation of relative growth analyses of the clam Tivela mactroides (Bivalvia, Veneridae)"
Authors: Alexander Turra, Guilherme N. Corte, A. Cecília Z. Amaral, Leonardo Q. Yokoyama, Márcia R. Denadai

GENERAL EVALUATION:

After another careful revision of the manuscript, I can confirm that the authors adopted most changes and improvements suggested by the reviewers in the previous evaluation of the manuscript. Accordingly, I recommend the acceptance of the manuscript for publication in PeerJ following “minor revision” (please see the specific / detailed comments provided below).

Specific comments:

Running head:
- line 13: I suggest replacing “Allometric growth of Tivela mactroides” by “Relative growth in Tivela mactroides”.

Abstract:
- line 22: Please replace “have been suggesting” by “have suggested”.
- line 24: Please replace “different models” by “diverse models”.
- line 31: Please replace “soft part weight parameters” by “soft parts weight parameters”.

Introduction:
- line 46: Please replace “Vasconcelos et al. 2016” by “Vasconcelos et al. 2018”.
- line 53 (and throughout the entire manuscript): Please replace “e.g.,” by “e.g.”.
- line 54: Please replace “Vasconcelos et al. 2016” by “Vasconcelos et al. 2018”.
- line 59: Please replace “Vasconcelos et al. 2016” by “Vasconcelos et al. 2018”.
- line 64 (and throughout the entire manuscript): Please replace “i.e.,” by “i.e.”.
- line 66: Please remove: “, for example”.
- line 67: Please replace “models which consider” by “models that consider”.
- line 75: Please replace “Vasconcelos et al. 2016” by “Vasconcelos et al. 2018”.
- line 79: Please replace “patterns on the shell” by “patterns of the shell”.

Materials and Methods:
- line 110: Please replace “16 Km” by “16 km”.
- line 115: Please replace “sea temperature” by “seawater temperature”.
- line 117: Please replace “variations are only found near the rivers” by “variations only occur near the rivers”.
- line 122: Please replace “through October 2004” by “until October 2004”.
- line 123: Please replace “Fig 1” by “Fig. 1”.
- line 127: Please replace “At the intertidal zone” by “In the intertidal zone”.
- line 131: Please replace “at the intertidal area” by “in the intertidal area”.
- line 135: Please replace “30 samples (dredgings) were collected per month” by “30 samples were dredged per month”.
- line 139: Please replace “had their shell length measured” by “measured for shell length”.
- line 146: Please replace “the soft parts dry weight and the shell weight” by “the shell weight and the soft parts dry weight”.
- line 148: Please replace “(Vasconcelos et al. 2008; 2016)” by “(Vasconcelos et al. 2008, 2018)”.
- line 169: Please replace “estimeted” by “estimated”.
- line 170: Please replace “rather than being from only one” by “rather than from one”.

Results:
- line 199: Please replace “relations” by “relationship”.
- line 204: Please replace “The allometric growth” by “The relative growth”.
- line 206: Please replace “allometric growth between shell length and width was always positive” by “relative growth between shell length and width was always positive allometric”.
- lines 207-208: Please replace “20 mm in length,” by “20 mm in L,”.
- line 208: Please replace “show that” by “showed that”.
- line 209: Please replace “; however, a rapid” by “followed by a rapid”.
- line 214: Please replace “observed” by “established”.

Discussion:
- line 225: Please replace “show that” by “showed that”.
- line 242: Please replace “allometric regressions models” by “allometric regression models”.
- line 253: Please replace “sea temperature” by “seawater temperature”.
- line 254: Please replace “also contributes” by “probably also contributes”.
- line 258: Please replace “between shell measure parameters” by “between shell parameters”.
- line 259: Please replace “in the morphometric relationships” by “in morphometric relationships”.
- line 260: Please replace “Alterations” by “Changes”.
- line 261: Please replace “normally” by “usually”.
- line 268: Please replace “which shows” by “that shows”.
- lines 279-280: Please replace “burrowing in this sediment.” by “burrowing in this type of sediment.”.
- line 283: Please replace “Besides bringing” by “Besides providing”.
- line 285: Please replace “relationships between soft parts and shell” by “relationships between shell size and soft parts weight”.
- line 289: Please replace “was observed” by “was recorded”.
- lines 289-290: Please replace “should target individuals with shell length between 25-30 mm preferably.” by “should preferably target individuals with shell length between 25-30 mm.”.
- line 291: Please replace “, and also provide” by “, thus providing”.

Conclusions:
- line 300: Please replace “recommendations” by “recommendation”.

References:
- lines 436-439: Please replace:
“Vasconcelos P, Moura P, Pereira F, Pereira AM, and Gaspar MB. 2016. Morphometric relationships and relative growth of 20 uncommon bivalve species from the Algarve coast (southern Portugal). Journal of the Marine Biological Association of the United Kingdom:1-12.
10.1017/S002531541600165X”
By:
“Vasconcelos P, Moura P, Pereira F, Pereira AM, and Gaspar MB. 2018. Morphometric relationships and relative growth of 20 uncommon bivalve species from the Algarve coast (southern Portugal). Journal of the Marine Biological Association of the United Kingdom:98(3):463-474.
https://doi.org/10.1017/S002531541600165X

Tables:
- Table 2 - line 6: Please replace “suitable model” by “suitable models”.

Figures:
- Figure 3 - Legend: Please replace “describing the relationship” by “describing the morphometric relationship”.
- Figure 3 - Models equations and descriptive statistics: Please use 3 decimal places in all “b” and “SE” values (please check figures c, d, e and f).

Annotated reviews are not available for download in order to protect the identity of reviewers who chose to remain anonymous.

---

## Round 0.3 · accepted · Accept

· Academic Editor

Accept

Thank you for your kind words regarding the PeerJ review process of your manuscript. For me it was also a pleasure to have the opportunity to deal with authors as yourself.

#